# A Differentiable Loss Function for Learning Heuristics in A*

## Abstract

Optimization of heuristic functions for the A* algorithm, realized by deep neural networks, is usually done by minimizing square root loss of estimate of the cost to goal values. This paper argues that this does not necessarily lead to a faster search of A* algorithm since its execution relies on relative values instead of absolute ones. As a mitigation, we propose the $L^*$ loss, which upper-bounds the number of excessively expanded states inside the A* search. The $L^*$ loss, when used in the optimization of state-of-the-art deep neural networks for automated planning in maze domains like Sokoban and maze with teleports, significantly improves the fraction of solved problems, the quality of founded plans, and reduces the number of expanded states to approximately 50%.

## 1 Introduction

Automated planning aims to find a sequence of actions that will reach a goal in a model of the environment provided by the user. Planning is considered to be one of the core problems in Artificial intelligence and it is behind some of its successful applications Samuel (1967); Knuth & Moore (1975); Silver et al. (2017). Early analysis of planning tasks McDermott (1996) indicated that optimising the heuristic function steering the search for a given problem domain can dramatically improve the performance of the search.

Learning in planning means optimizing heuristic functions from plans of already solved problems and their instances. This definition includes selection of proper heuristics in a set of pattern databases Franco et al. (2017); Haslum et al. (2007); Moraru et al. (2019); Edelkamp (2006), a selection of a planner from a portfolio Katz et al. (2018), learning planning operators from instances Ménager et al. (2018); Wang (1994), and learning for macro-operators and entanglements Chrpa (2010); Korf (1985). Recent years observe a renewed interest in learning heuristic functions and this is fuelled by the success of deep learning and reinforcement learning in the same area Shen et al. (2020); Groshev et al. (2018); Ferber et al. (2020); Bhardwaj et al. (2017).

In this work, we are interested in optimising the heuristic function for A* Hart et al. (1968), which despite the popularity of Monte Carlo tree search Coulom (2006); Silver et al. (2017) is interesting due to its guarantees on optimal solution. A* is also optimally efficient in the sense that it expands the minimal number of states. Majority of prior art Shen et al. (2020); Toyer et al. (2020); Groshev et al. (2018); Ferber et al. (2020); Bhardwaj et al. (2017) optimises the heuristic function by minimizing the error of the predicted cost to the goal on a *training set* of problem instances,[1] where the error is measured by the $L_2$ error function or its variant. $L_2 = 0$ does not guarantee the optimal efficiency of A*, hence it gives a false sense of security.

We propose a $L^*$ loss function tailored for A*, which minimizes an upper bound on the number of expanded states. This is achieved by stimulating states on an optimal path to have a smaller cost function $f = g + h$ than those off the optimal path. By this, $L^*$ effectively utilizes all the states generated during the exploration of A*, providing much more information to the learner. If $L^*$ on a given problem instance is equal to zero, it is guaranteed that A* will expand only states on the optimal path, which under conditions on the training set as detailed below, implies optimal efficiency of A*. We emphasize that the optimal efficiency is retained even on problems with

---

[1]The training set contains solved problem instances, where the solution should be ideally found by a search finding optimal solution, such as A* with ideally admissible heuristic function.

exponentially many optimal paths Helmert & Röger (2008), therefore the heuristic function has to learn a tie-breaking mechanism.

The proposed L* is compared to state of the art on seven domains: Sokoban, Maze with teleports, Sliding tile puzzle, Blockworld, Ferry, Grippe, and N-Puzzle and on all of them it consistently outperforms heuristic functions optimizing $L_2$.

## 2 PRELIMINARIES

We define a search problem instance by a directed weighted graph $\Gamma = \langle S, \mathcal{E}, w \rangle$, a distinct node $s_0 \in S$ and a distinct set of nodes $S^* \subseteq S$. The nodes $S$ denote all possible states $s \in S$ of the underlying transition system representing the graph. The set of edges $\mathcal{E}$ contains all possible transitions $e \in \mathcal{E}$ between the states in the form $e = (s, s')$. $s_0 \in S$ is the initial state of the problem instance and $S^* \subseteq S$ is a set of allowed goal states. Problem instance graph weights (alias action costs) are mappings $w : \mathcal{E} \to \mathbb{R}^{\geq 0}$.

Let $\pi = (e_1, e_2, \ldots, e_l)$, we call $\pi$ a path (alias a plan) of length $l$ solving a task $\Gamma$ with $s_0$ and $S^*$ iff $\pi = ((s_0, s_1), (s_1, s_2), \ldots, (s_{l-1}, s_l))$ and $s_l \in S^*$. An optimal path is defined as a minimal cost of a problem instance $\Gamma, s_0, S^*$ and is denoted as $\pi^*$ together with its value $f^* = w(\pi^*) = w(e_1) + w(e_2) + \ldots, + w(e_l)$. We often minimize the cost of solution of a problem instance $\Gamma, s_0, S^*$, namely $\pi^*$, together with its length $l^* = |\pi^*|$.

### 2.1 A* ALGORITHM

Let's briefly recall how the A* algorithm works. For consistent heuristics, where $h(s) - h(s') \leq w(s, s')$ for all edges $(s, s')$ in the $w$-weighted state space graph, it mimics the working of Dijkstra's shortest-path algorithm Dijkstra (1959) and maintains the set of generated but not expanded nodes in $\mathcal{O}$ (the Open list) and the set of already expanded nodes in $\mathcal{C}$ (the Closed list). It works as follows.

1. Add the start node $s_0$ to the Open list $\mathcal{O}_0$.
2. Set $g(s_0) = 0$
3. Initiate the Closed list to empty, i.e. $\mathcal{C}_0 = \emptyset$.
4. For $i \in 1, \ldots$ until $\mathcal{O}_i \neq \emptyset$
   (a) Select the state $s_i = \arg\min_{s \in \mathcal{O}_{i-1}} g(s) + h(s)$
   (b) Remove $s_i$ from $\mathcal{O}_{i-1}$, $\mathcal{O}_i = \mathcal{O}_{i-1} \setminus \{s_i\}$
   (c) If $s_i \in S^*$, i.e. it is a goal state, go to 5.
   (d) Insert the state $s_i$ to $\mathcal{C}_{i-1}$, $\mathcal{C}_i = \mathcal{C}_{i-1} \cup \{s_i\}$
   (e) Expand the state $s_i$ into states $s'$ for which hold $(s_i, s') \in \mathcal{E}$ and for each
      i. set $g(s') = g(s_i) + w(s_i, s')$
      ii. if $s'$ is in the Closed list as $s_c$ and $g(s') < g(s_c)$ then $s_c$ is reopened (i.e., moved from the Closed to the Open list), else continue with (e)
      iii. if $s'$ is in the Open list as $s_o$ and $g(s') < g(s_o)$ then $s_o$ is updated (i.e., removed from the Open list and re-added in next step with updated $g(\cdot)$), else continue with (e)
      iv. add $s'$ into the Open list
5. Walk back to retrieve the optimal path.

In the above algorithm, $g(s)$ denotes a function assigning an accumulated cost $w$ for moving from the initial state ($s_0$) to a given state $s$. Consistent heuristics are called monotone because the estimated cost of a partial solution $f(s) = g(s) + h(s)$ is monotonically non-decreasing along the best path to the goal. More than this, $f$ is monotone on all edges $(s, s')$, if and only if $h$ is consistent as we have $f(s') = g(s') + h(s') \geq g(s) + w(s, s') + h(s) - w(s, s') = f(s)$ and $h(s) - h(s') = f(s) - g(s) - (f(s') - g(s')) = f(s) - f(s') + w(s, s') \leq w(s, s')$. For the case of consistent heuristics, no reopening (moving back nodes from Closed to Open) is needed, as we essentially traverse a state-space graph with edge weights $w(s, s') + h(s') - h(s) \geq 0$. For the trivial heuristic $h_0$, we have $h_0(s) = 0$ and for perfect heuristic $h^*$, we have $f(s) = f^* = g(s) + h^*(s)$ for all nodes $s$. Both heuristics $h_0$ and $h^*$ are consistent.

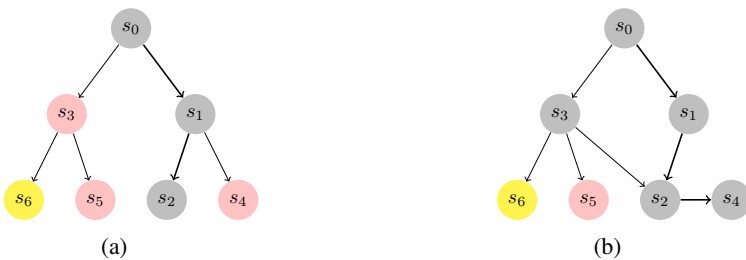

(a)                                                                 (b)

Figure 1: A visualization of a search space of an A* algorithm. In sub-figure (a), path $s_0 \rightarrow s_1 \rightarrow s_2$ represents the optimal plan, states $\{s_3, s_4, s_5\}/s_6$ are off the optimal path but have / have not been generated by the A* . In sub-figure (b), path $s_0 \rightarrow s_1 \rightarrow s_2 \rightarrow s_4$ and $s_0 \rightarrow s_1 \rightarrow s_3 \rightarrow s_4$ represents the optimal plan, states $s_5/s_6$ are off the optimal path but have / have not been generated.

Even if the heuristic is not consistent, algorithms like A* even without the reopening, remain complete i.e. they find a plan if there is one. Plans might not be provably optimal but are often very good in planning practice.

## 2.2 Optimizing the heuristic

We consider heuristic function $h_\theta : \mathcal{S} \rightarrow \mathbb{R}^{\geq 0}$ mapping a state $s \in \mathcal{S}$ to a real non-negative value, where $\theta \in \mathbb{R}^m$ holds parameters of $h_\theta$. Using a set of problem instances $\mathcal{T}$ (further called training set), we want to optimize parameters $\theta$ of $h_\theta$ such that an A* search algorithm would find an (optimal) solution by expanding the least number of states. [2] This, in practice, means to solve the optimization problem

$$\arg\min_\theta \sum_{\mathcal{S} \in \mathcal{T}} \mathrm{L}(h_\theta, \mathcal{S}), \tag{1}$$

where the loss function L should be such that smaller values imply better heuristic function $h_\theta$ as perceived by A* .

## 2.3 Weakness of $L_2$ loss function

Many prior art on optimizing heuristic function Shen et al. (2020); Groshev et al. (2018); Ferber et al. (2020); Bhardwaj et al. (2017); Toyer et al. (2020) minimize the $L_2$ loss function[3] $L_2(h_\theta, \mathcal{S}) = \sum_i \left( h_\theta(s_i) - y_i \right)^2$ , where the training set $\mathcal{S}$ consists of pairs $\{(s_i, y_i)\}_{i=1}^n$ , where $s_i$ is some state and $y_i$ is the length of the plan from $s_i$ to the goal state. We argue that zero $L_2$ loss on a given *problem instance* for states on the optimal path does not guarantee that *A* will be optimally efficient* in the sense that it can expand more states than needed.

**L2 does not utilize states off the optimal path.** Imagine a problem instance shown in Figure 1a, where $s_0$ and $s_2$ are the initial and goal states respectively and $(s_0, s_1, s_2)$ is the optimal path. When L2 loss is optimized, one needs to know the exact cost-to-go values which are obtained by solving the problem instance. Thus, by solving the instance in Figure 1, one obtains heuristic values for $\{s_0, s_1, s_2\}$. But if $L_2$ loss is equal to zero on them, it does not say anything about estimates for states off the optimal path states $\{s_3, s_4, s_5, s_6\}$. This means that it can happen that $f(s_3) < f(s_1)$, which would lead to expanding the state $s_3$ in A* algorithm and hence to sub-optimality.

This issue can be fixed, if the training set is extended to contain heuristic values for all states off the optimal path ($\{s_3, s_4, \ldots, s_6\}$ in our example), which in practice requires solving all possible variants of the problem instances. This has been suggested in Bhardwaj et al. (2017) but is infeasible due to excessive computational requirements. Therefore, in practice, it is assumed the training set to be large, thereby mitigating this problem.

---

[2]Here it is assumed that the number of expanded states is directly proportional to the time taken to find the solution, as the time to compute the value of $h_\theta$ is independent of the value of $\theta$.

[3]While some works use $L_1$, the properties discussed here for $L_2$ holds for $L_1$ as well.

$L_2$ **loss provides a false sense of optimality.** Some problems can have large (even exponential) number of optimal solutions with the same cost Helmert & Röger (2008). In this case, minimizing estimate of cost-to-go of all states in the problem instance (to fix the problem mentioned above) does not guarantee that A* will be optimally efficient. Consider an example in Figure 1b with unit cost on edges. The algorithm starts by expanding state $s_0$ to $s_1$ and $s_2$. The heuristic is perfectly estimated and $f(s_1) = f(s_2) = 3$. Hence there are two states with the same value $f$ which means A* has to decide, how to break this tie. The situation repeats after A* expands either $s_1$ or $s_3$, since the open set will now contain state $s_3$ with $f(s_3) = 3$ and A* needs to resolve ties again. See Helmert & Röger (2008) for more examples.

**Heuristic value for unreachable (dead-end) states** should be infinite to ensure that they are never selected. An infinity in the $L_2$ loss would always lead to an infinite loss which would then result in an infinite gradient. Hence, in practice, a sufficiently large value for dead-end states has to be used.

## 3 L* LOSS

We explain the proposed L*loss function on a single problem instance $\Gamma = \langle \mathcal{S}, \mathcal{E}, w \rangle$ (the extension to a set of plan is trivial through Equation equation 1). We assume to have a (preferably optimal and shortest) plan $\pi = ((s_0, s_1), (s_1, s_2), \ldots, (s_{n-1}, s_n))$ with states from this optimal plan denoted as $\mathcal{S}^o = \{s_0, s_1, s_2, \ldots, s_n\}$. This plan can be found by A* with some (admissible) heuristic function $h$, which *does not have to* coincide with the heuristic function $h_\theta$ that we are optimizing. We denote states off the optimal plan as $\mathcal{S}^n \subset \mathcal{S} \setminus \mathcal{S}^o$, where the subset exists because, in practice, $\mathcal{S}^n$ contains states generated by A* while solving the problem instance $\Gamma$. In the visualization in Figure 1a, grey states are on the optimal path $\mathcal{S}^o$, pink states are off the optimal path, and yellow states were not generated in the course of solving the problem instance. Hence, $\mathcal{S} = \{s_i\}_{i=1}^6$, $\mathcal{S}^o = \{s_i\}_{i=1}^2$ and $\mathcal{S}^n = \{s_i\}_{i=3}^5$. The training sample for a L*is defined as a tuple $\bar{\Gamma} = \langle \mathcal{S}, \mathcal{E}, w \rangle, \mathcal{S}^o$.

L* aims to minimize the number of expanded states in the A* algorithm. Recall that A* always expands a state from an open list with smallest $f_\theta(s) = g(s) + h_\theta(s)$. To be optimally efficient, states on the optimal path $s' \in \mathcal{S}^o$ should have *always* smaller $f(s')$ than states off the optimal path $s'' \in \mathcal{S}^n$, i.e.

$$(\forall s' \in \mathcal{S}^o)(\forall s'' \in \mathcal{S}^n)(g(s') + h_\theta(s') < g(s'') + h_\theta(s'')) \qquad (2)$$

On the optimal path, we might also impose monotonicity as

$$(\forall s_i, s_j \in \mathcal{S}^o)(i < j)(g(s_i) + h_\theta(s_i) \le g(s_j) + h_\theta(s_j), \qquad (3)$$

though it does not affect the optimality of A* . We do this, since monotonic heuristic function implies A* returning optimal solution. In Constraint equation 2, states not generated by A* are ignored. But $\mathcal{S}^n$ will always contain all states of distance one from the optimal path, which is sufficient to show that a loss equalling zero implies expanding states only on an optimal path (in the training set). To prevent confusion, we emphasize that conditions are designed for the heuristic $h_\theta$ that is to be optimized, and not for the heuristic $h$ that has generated the training set in the first place.

While Constraint equation 3 is true for every consistent heuristic, Constraint equation 2 is true only for perfect heuristics. Otherwise, we could have some earlier states in the exploration off the optimal path that have a smaller $f$-value than later ones in the optimal path. What seems to be over-restrictive, such that almost no heuristic function will ever fulfill, Constraint equation 2, is intentional.

The proposed L* loss minimizes the number of times each of the above constraints are violated as

$$\frac{1}{|\mathcal{S}^o||\mathcal{S}^n|} \sum_{s' \in \mathcal{S}^o} \sum_{s'' \in \mathcal{S}^n} [\![g(s') + h_\theta(s') \ge g(s'') + h_\theta(s'')]\!] +$$

$$\frac{1}{|\mathcal{S}^o|(|\mathcal{S}^o| - 1)} \sum_{i=2}^{|\mathcal{S}^o|} \sum_{j=1}^{i} [\![g(s_i) + h_\theta(s_i) > g(s_j) + h_\theta(s_j)]\!], \qquad (4)$$

where $[\![\cdot]\!]$ is an Iverson bracket, which is equal to one if the argument is true and zero otherwise. The first part of the loss function loosely upper bounds the number of non-optimal states the A* expands while the second part ensures the monotonicity of the heuristic function along the optimal plan. In other words, the conditions equation 2 and equation 3 encode the aim of a consistent and

perfect heuristic. During training, we iterate over many samples of A* explorations which enlarges the scope of $L^*$.

To set constraints for heuristic learning, we only need the partitioning of the set of explored nodes into the sets $S^0$ and $S^n$, computed via an optimal plan and a set of all generated nodes, together with their $g$-values. Given the optimal heuristic, A* will always find an optimal solution. Up to tie-breaking, it is optimally efficient and will expand only nodes with optimal merit $f^*$.

Loss function $L^*$ does not distinguish between the Open and Closed lists in the exploration of A* as long as it has access to the combined set of explored nodes. This way, we can take any optimal planner and not just the heuristic search planners for training.

### 3.1 How $L^*$ addresses the deficiencies of $L_2$

$L^*$ **utilizes all states** generated during the A* search used to create the training sample(s), which is in sharp contrast to $L_2$ estimating cost-to-go. This propagates to better utilization of states in the training set. The experimental results show that given a fixed and small number of training problems, models minimizing $L^*$ achieves higher performance.

$L^* = 0$ **implies optimality.** We state a following theorem:

**Theorem 1 (Upper Bound)** *For a problem instance with states $\mathcal{S} = \mathcal{S}^n \cup \mathcal{S}^o$, denote*

$$\mathcal{R}^n = \left\{ s'' \in \mathcal{S}^n \mid \exists s' \in \mathcal{S}^0 \wedge (g(s') + h_\theta(s') \geq g(s'') + h_\theta(s'')) \right\}, \tag{5}$$

*the quantity $|\mathcal{R}^n|$ is an upper bound on the number of non-optimal states A* expands during its search.*

The proof is straightforward and it is included in supplementary for completeness. The quantity $|\mathcal{R}^n|$ is exactly the quantity minimized by the $L^*$ as defined in Equation equation 4. The following theorem is a trivial consequence of this property.

**Theorem 2 (Optimal efficiency)** *Let for a given training sample $\bar{\Gamma}$, and a heuristic function $h_\theta$ $L^*(\Gamma, h_\theta) = 0$. Then A* with heuristic function $h_\theta$ will expand only states on the optimal path $\mathcal{S}^o$. If $\mathcal{S}^o$ in $\bar{\Gamma}$ is optimal and shortest, A* will be optimally efficient.*

The proof is a consequence of the property that $L^*(\Gamma, \bar{h_\theta}) = 0$ implies that $|\mathcal{R}^n| = 0$. The above theorem holds even on problems with multiple optimal solutions. In this case, $L^*$ would be either equal to zero, which means $h_\theta$ includes tie-breaking mechanism and it will be optimal, or it will be greater than zero. Thus and unlike $L_2$, its zero value implies optimal efficiency.

$L^*$**does not require heuristic value of unreachable (dead-end) states,** which is caused by the fact that $L^*$requires satisfaction of inequalities instead of estimation of some value. $L^*$ loss does not force the heuristic to be goal aware, since as discussed in Supplementary, this is not needed for the optimal efficiency of A* .

## 4 Related Work

In potential heuristics Seipp et al. (2015), parameters of the heuristic functions are optimized by linear programming for a particular problem instance to satisfy constraints similar to those stated in this paper. The optimization assumes a particular structure of the heuristic, unlike here, where no structure is assumed. Ref. Takahashi et al. (2019) admits that the symmetry of $L_2$ (and of $L_1$) loss does not promote admissibility of the heuristic. It recommends asymmetric $L_1$ with different weights on the left and right parts, but this does not completely solve the problems identified above in Section 2.3.

In Ferber et al. (2020), neural networks estimate the number of expansions of a GBFS search, though the results are comparable to an estimation of cost-to-goal. In Bhardwaj et al. (2017) A* is viewed as a Markov Decision Process with value function being equal to the number of steps of A* till it reaches the solution. While this detaches the heuristic values from cost-to-goal cost, it does not solve the problem with dead-ends, state efficiency, and ties.

Refs. Vlastelica et al. (2021); Yonetani et al. (2021) combine neural networks with discrete search algorithms, which become an inseparable part of the architecture. Orseau & Lelis (2021) proposes a new search algorithm that uses both a heuristic function and policy networks to minimize the search loss. Our setting is more classical where the heuristic is optimized for A* search but the execution of the search is independent of the training. This has the advantage that one (costly) execution of A* search algorithm is used many times during training to optimize weights.

A large corpus of literature Silver et al. (2017); Guez et al. (2018); Feng et al. (2020); Anthony et al. (2017) is devoted to improvements to Monte Carlo Tree Search. Since this work is concerned exclusively to A* algorithm, we view these works independent to this. Nevertheless, we compare to some of them in the experimental section. Similarly, a lot of works Shen et al. (2020); Toyer et al. (2020); Zhang & Geißer (2021); Groshev et al. (2018); Chrestien et al. (2021); Ferber et al. (2020); Bhardwaj et al. (2017) investigate architectures of neural networks for learning a heuristic function, ideally for arbitrary planning problems. These works are perpendicular to this one, which investigate how to optimize these neural networks to perform well inside A* .

## 5 EXPERIMENTAL EVALUATION

Heuristic functions optimized with respect to $L^*$ loss function are compared to those optimized with respect to $L_2$ loss on seven domains. Due to lack of space, only Sokoban and Mazes with teleports are detailed below, and the remaining five are in supplementary material. This is supplemented by the comparison with domain-independent planners: (1) SymBA* Torralba et al. (2014), a cost-optimal planner from International Planning Competition (IPC) 2014; (2) Delfi Katz et al. (2018); (3) Mercury14 Katz & Hoffmann (2014), a satisficing planner from IPC 2014; (4) Stone soupe Seipp & Röger (2018), and by solutions based on Monte Carlo Tree Search Guez et al. (2018) and reinforcement learning Racanière et al. (2017); Guez et al. (2019). Ref. Toyer et al. (2020) admit it does not work on Sokoban and Shen et al. (2020) works only on small Sokoban problems with two boxes, we do not compare to these works. All experiments involving neural networks have been repeated three times.

**The Neural Network**  Neural networks (NN) implementing heuristic functions were adopted from Groshev et al. (2018) and Chrestien et al. (2021), where the latter is, to our best knowledge the state-of-the-art architecture for maze domains. It contains seven convolution layers $P_1, \ldots P_7$ followed by four convolution-attention-position blocks, which allow correlating information from distant parts of the maze. The output tensor of the fourth CoAt block is "flattened" by global average pooling over the $x$ and $y$ dimension to a vector, which is then fed to a fully connected layer (FC) which outputs a scalar estimating the heuristic value. More details on the network can be found Chrestien et al. (2021). This network is further called "CoAt" after the CoAt blocks. We also studied the network of Groshev et al. (2018) (without policy head), which has a similar structure, but instead of four CoAt blocks it has seven CNN layers. We refer to this network as to CNN. Both networks are by design scale-free, which means that they can be used on mazes of various sizes, as is shown below on the mazes with teleport domain. Since the $L^*$loss as defined in Equation 4 is not differentiable, the Iverson bracket is replaced by a logistic loss function $L_l(x) = \log(1 + \exp(-x))$.

Our experiments were implemented in Keras-2.4.3 with Tensorflow-2.3.1 as the backend. For training the neural networks, we used an NVIDIA Tesla GPU model V100-SXM2-32GB; the evaluation was performed on the CPU to ensure a fair comparison to domain independent planners. The neural networks were trained by the Adam optimizer Kingma & Ba (2014) with a default learning rate of 0.001. Each mini-batch contained **all** states from one problem instance. Scripts reproducing our experiments together with mazes and solutions will be made available upon acceptance.

### 5.1 TRAINING FROM SOLVED MAZES

We generate mazes for the training set by running the A* algorithm using the heuristic function from Groshev et al. (2018) on a set of problem instances to identify a set of states generated during the A* search. All these sets form the training set. Since optimizing the heuristic by $L_2$ loss requires knowing the true heuristic value (cost to reach the goal), we have used SymBA* Torralba et al. (2014) to find the optimal plan from each state in the training set. For states for which SymBA* doesn't find a solution (dead-end states), the $h$ value is replaced by a very large value. This con-

|  |  |  |  |  | CNN | | CoAt | | |
|---|---|---|---|---|---|---|---|---|---|
| #b | SBA* | Delfi | Merc | FDSS | $L_2$ | $L^*$ | $L_2$ | $L^*$ | CL w.$L^*$ |
| 3 | 100 | 100 | 100 | 100 | 81 | 87 | 91 | 94 | 95 |
| 4 | 100 | 100 | 81 | 100 | 74 | 80 | 89 | 93 | 94 |
| 5 | 97 | 91 | 67 | 94 | 72 | 82 | 85 | 89 | 90 |
| 6 | 55 | 55 | 49 | 56 | 61 | 71 | 73 | 80 | 85 |
| 7 | 46 | 44 | 31 | 42 | 51 | 59 | 63 | 77 | 83 |
| 8 | - | - | - | - | - | - | - | 32 | 59 |
| 9 | - | - | - | - | - | - | - | 12 | 38 |

(a)

| model | coverage |
|---|---|
| MCTSNet | 84 |
| I2A | 84 |
| DRC (3,3) 10k | 93 |
| DRC (3,3) 900k | 99 |
| $CoAt - L^*$ | 97 |
| CL $CoAt - L^*$ | 100 |
| SBA* | 100 |

(b)

Table 1: **Left:** Fraction of solved mazes (in percents) of S(ym)BA* , Delfi(1), Merc(ury14), FDSS (Fast Downward Stone Soup), CoAt and CoAt* on test data sets containing variable number of boxes. Column captioned #b indicates the number of boxes in different categories. The standard deviation of all repeated experiments was between 0.004 and 0.008 and it is not shown to save space. **Right:** Fraction of solved mazes (in percents) from Boxoban dataset. DRC 900k / DRC 10k optimizes on $10k$ levels. Results of MCTSNet Guez et al. (2018), I2A Racanière et al. (2017), and DRC Guez et al. (2019) have been copied from Table 2 of Guez et al. (2019).

struction, albeit very expensive, allows a fair comparison, since the training of heuristic by $L_2$ loss will also use states off the optimal path of the original problem instance for which the states were generated.

**Sokoban**'s training set contained 10000 Sokoban mazes of size $10 \times 10$ with 3 boxes created using gym-sokoban Schrader (2018). The testing set contained 2000 mazes of the same size $10 \times 10$ but with $3, 4, 5, 6, 7, 8, 9$ boxes. The complexity increases with the addition of more boxes.[4], and therefore we can evaluate the ability to generalize outside training environments. We go a step further and implement curriculum learning Bengio et al. (2009) by training from those mazes that are solved by our network during evaluation. We create a new training set containing all the solved mazes and re-train our network in an effort to improve the coverage of our network.

**Maze-with-teleports**'s training set contained 5000 randomly generated mazes of size $n \times n = 15 \times 15$ with the agent in the upper-left corner and the goal in the lower-right corner. The mazes were generated using an open-source maze generator [5], where walls were randomly broken and 4 pairs of teleports were randomly added to the maze structure. The testing contained 2000 mazes generated by the same algorithm but (i) were bigger by up to $60 \times 60$ and (ii) were rotated by 90, 180, and 270 degrees which moved the start and goal states to positions not occurring in the training set.

## 5.2 RESULTS

**Sokoban**    Table 1a shows the percentage of solved mazes of all compared planners on problem instances with a various number of boxes (recall that the NNs were optimized on instances with only three boxes). All planners were given a time limit of 10 minutes to solve each Sokoban instance. On mazes with 3 and 4 boxes, the optimal planners SymBA* (SBA* ) and Delfi were able to solve all problem instances while the best performing architecture among the NNs, which is CoAt optimized with respect to $L^*$ ($CoAt - L^*$), could solve 94% and 93% of the mazes respectively. On increasing the number of boxes, the A* with NNs start outperforming classical planners. A* with NNs optimizing $L^*$ is consistently better than those optimizing $L_2$. The CoAt architecture proposed in Chrestien et al. (2021) with the proposed $L^*$ is the only solver that can solve some mazes with 8 and 9 boxes. Solutions found by CoAt optimizing $L^*$ were close to optimum, on average by 1 step longer, which is likely because the learnt heuristic is most of the times admissible (see Supplementary for details). The average number of expanded states in Figure 2a shows that NNs optimizing $L^*$ indeed expand a smaller number of states than those optimizing $L_2$.

Since networks have never seen mazes with four or more boxes, extrapolation to eight and nine boxes is impressive. To evaluate the potential for self-improvement, the training set of $CoAt - L^*$ was

---

[4]This is of course just an approximation, as we can have simple problems with a large number of boxes
[5]https://github.com/ravenkls/Maze-Generator-and-Solver

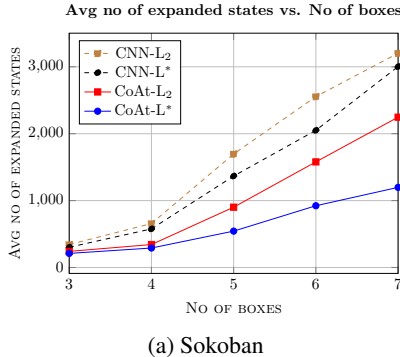

(a) Sokoban

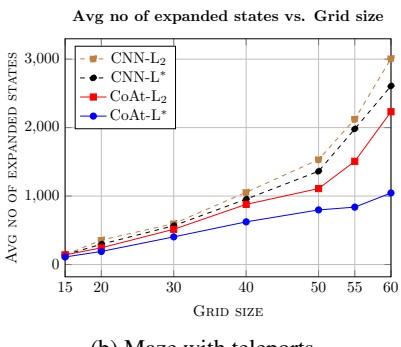

(b) Maze with teleports

Figure 2: The average number of expanded states in A* for Sokoban (left) and Maze problems (right).

| | | | | | CNN | | CoAt | |
|---|---|---|---|---|---|---|---|---|
| $n$ | SBA* | Delfi | Merc | FDSS | $L_2$ | $L^*$ | $L_2$ | $L^*$ |
| 50 | 92 | 90 | 75 | 100 | 100 | 100 | 100 | 100 |
| 55 | 52 | 50 | - | 100 | 85 | 85 | 86 | 88 |
| 60 | - | - | - | 100 | 73 | 74 | 76 | 79 |

Table 2: Fraction of solved mazes with teleports of S(ym)BA* , Delfi(1), Merc(ury14), and A* algorithm with heuristic function implemented by CNN and CoAt networks optimized with respect to $L_2$ and $L^*$. All solvers have solved all mazes of size 15–40, hence they are not shown Standard deviations of repeated experiments were between 0.005 and 0.008 are shown in Supplementary.

extended with mazes from the testing set it has already solved for fine-tuning. We refer to this as curriculum learning (CL w. $L^*$) and the results are shown in the last column of Table 1a. It shows marginal improvement on mazes with 3-5 boxes but records a significant improvement in performance over the vanilla $L^*$on mazes exceeding 5 boxes.

**Boxoban** On unfiltered "Boxoban" levels from Guez et al. (2019), A* with heuristic implemented by CoAt network and optimized with respect to the proposed $L^*$is compared to MCTSNet Guez et al. (2018), Imagination augmented agent (I2A) Racanière et al. (2017), and to DRC (3,3) network Guez et al. (2019), with possible discrepancies, as results on the competing methods were taken from Guez et al. (2019). $CoAt-L^*$was optimized on 10k mazes with 3 boxes (it is the same network as reported in the previous paragraph), whereas others were optimized on 900k mazes with 4 boxes. $CoAt-L^*$and MCTSNet knows the model, whereas DRC and I2A do not. The fraction of solved mazes shown in Table 1b shows that $CoAt-L^*$trained on 10k mazes is second best behind DRC (3,3) that is trained on 900k mazes, but DRC (3,3) trained on 10k mazes is already inferior. $CoAt-L^*$with one iteration of curriculum learning where the training set is extended to contain previously solved mazes (from set used in previous paragraph) solves 100% of boxoban mazes. Needless to say that during optimization, DRC (3,3) used 1G iterations of SGD, MCTSNets allowed 10M iterations of SGD, whereas CoAt optimized $L^*$allowed just for 120k iterations, which is few magnitudes less.

**Maze-with-teleports** Fraction of solved mazes with teleports is shown in Table 2. A* with heuristic implemented by NN was optimized on mazes of size $15 \times 15$ and has solved all mazes up to size $40 \times 40$ (not shown in the table) and beyond. The results mimic the results on Sokoban in the sense that A* with CoAt networks optimizing $L^*$ is consistently outperforming those optimizing $L_2$.

All 2000 mazes in the training set were created such that the agent starts in the top left corner and the goal is in the bottom right corner. When mazes are rotated by 90°, 180° and 270°, the agent has to solve mazes with distributions very different to that on the training set, yet the fraction of solved mazes for $CoAt-L^*$decreases by at most 5% (see Table 3 in Supplementary). Average number of generated states in A* with different heuristics is shown in Figure 2a. Again, heuristics optimized with respect to $L^*$expand smaller number of states during the search.

| epoch | 3000 | | 4000 | | 5000 | | 6000 | |
|---|---|---|---|---|---|---|---|---|
| | $L_2$ | $L^*$ | $L_2$ | $L^*$ | $L_2$ | $L^*$ | $L_2$ | $L^*$ |
| 0 | 7.4 | 7.4 | 7.7 | 7.7 | 8.5 | 8.5 | 8.3 | 8.3 |
| 1 | 11 | 11 | 9.8 | 11 | 10 | 10 | 9.6 | 10 |
| 2 | 15 | 18 | 11 | 16 | 12 | 18 | 14 | 16 |
| 3 | 29 | 33 | 20 | 31 | 21 | 24 | 20 | 32 |
| 4 | 34 | 62 | 27 | 69 | 36 | 49 | 31 | 48 |
| 5 | 62 | 75 | 44 | 75 | 57 | 73 | 34 | 66 |
| 6 | 67 | 77 | 51 | 80 | 61 | 80 | 49 | 77 |
| 7 | 63 | 76 | 62 | 80 | 74 | 82 | 50 | 86 |

(a) Sokoban

| epoch | 1000 | | 3000 | | 5000 | |
|---|---|---|---|---|---|---|
| | $L_2$ | $L^*$ | $L_2$ | $L^*$ | $L_2$ | $L^*$ |
| 0 | 25 | 25 | 19 | 19 | 15 | 15 |
| 1 | 42 | 45 | 29 | 35 | 18 | 20 |
| 2 | 45 | 68 | 39 | 51 | 34 | 34 |
| 3 | 69 | 83 | 59 | 67 | 41 | 59 |
| 4 | 84 | 90 | 78 | 83 | 66 | 75 |

(b) Maze with teleports

Table 3: Fraction of solved mazes (in percents) when the networks are optimized only on mazes they have previously solved. First row corresponds to A* with untrained network.

### 5.3 TRAINING FROM UNSOLVED MAZES

Let's now consider a bootstrap protocol, where in each epoch, a heuristic function implemented by the NN is first used in A* to try to solve mazes from an available set of mazes (recall that we set a 10min time limit for solving a maze) and then to optimize its parameters on a set of mazes it has solved. Similarly to reinforcement learning, if an un-optimized (which means uninformed) heuristics solves at least a few mazes, it can boot the learning.

In this experiment, the training set of unsolved mazes is fixed. In the optimization of our NN over solved mazes, we perform one epoch. Hence the number of iteration and epoch coincide. Table 3 shows percentages of solved mazes on Sokoban and Maze with teleports for the first seven and four epochs (epoch number 0 means that the network is untrained) for different sizes of the training set. The set of Sokoban mazes contained problems with three, four, and five boxes; the set of mazes with teleports contained problems of size $40 \times 40$.

We observe that the fraction of solved mazes increases with epochs and the speed of this growth is significantly faster for heuristics optimized with respect to the proposed $L^*$. To our surprise, the fraction of solved mazes does not grow faster when the number of initially unsolved set of mazes is bigger. Yet we have observed that the fraction of solved unfiltered boxoban mazes increases as expected. A* with the network optimized on the set of 6000 mazes could solve 96% of levels of unfiltered boxoban mazes after seven epochs. This agent has performed just 20.5k gradient descend steps, which is comparatively smaller than 1G steps of DRC (3,3) agent from Guez et al. (2018). Sections 5 and 6 in Supplementary contains additional evaluation of five more domains (Sliding tile puzzle, Blockworld, Gripper, Ferry, and N-Puzzle) with the same conclusion: $L^*$ *is always better than* $L_2$.

## 6 CONCLUSION

This work has proposed $L^*$ loss function for imitation learning in planning;it has been designed specifically to maximize the efficiency of the A* algorithm. $L^*$ is zero, if and only if the basic monotonicity requirements on the f-value in A* are satisfied, so that the heuristic trained on this loss function is consistent and perfect. This enables A* to find the optimal solution at an optimal time. It has been shown that $L_2$ does not have these guarantees. The experiments have verified the promises that A* with heuristic functions optimized with respect to $L^*$ **always** solve a much higher number of problems, generate up to 50% lesser states than those optimized with respect to the usual $L_2$, and return nearly optimal solutions. By comparison to MCTSNets, we have shown that A* with well trained heuristic can be competitive to Monte Carlo Tree Search. The training is also more efficient, as we use a much lesser number of SGD steps.

The proposed $L^*$loss well complements contemporary research, which pays a lot of attention to the network architectures, as it can be used as a drop-in replacement for $L_2$. It inspires us to design loss functions for other types of search algorithm and research neglected aspects, such as the construction of a representative training set. We see them as a limiting factor in the further endeavour to solve more difficult problem instances.

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
