# OpenReview forum: "A Differentiable Loss Function for Learning Heuristics in A*"
_ICLR.cc/2023/Conference — Submitted to ICLR 2023_

### Official Review · Reviewer_ytHz · 2022-10-23

**Confidence:** 3
**Correctness:** 3
**Technical Novelty And Significance:** 4
**Empirical Novelty And Significance:** 4
**Recommendation:** 8

**Clarity, Quality, Novelty And Reproducibility:**

As mentioned under Strength And Weaknesses, there are some clarity issues. However, the gist of the paper is clear. Most importantly, the theoretical and the empirical results (and methodology) are clear.
The quality is good (great if listed issues are fixed), novelty is very high and reproducibility is high.

**Strength And Weaknesses:**

Strengths
==========

The theoretical investigation on $L_2$ limitations and the desired and acquired properties of $L$*

The evaluation is very strong with respect to the selected problem domains.
* $L_2$-loss is trained using the same training data as $L$*-loss (i.e. off the optimal path as well).
* The use of an automatic planning method which is optimal (S(ym)BA*) as baseline.
* Comparison with MCTS and model free methods DRC & I2A.
* Learning with curriculum learning from random initiation with only training data produced by the method itself.

Learning $L$* does not presuppose a distinction between open and closed sets. That is, the training data can be generated using other methods than search.


Weaknesses
==========

Some parts of the paper, including equations, are not clear.
* I can see no \cdot but I interpret the $leq$ expression as an implicit indicator function (equivalent to the use of an Iverson bracket). I do however strongly suggest that you make this clear in the equation. E.g. by introducing an Iverson bracket explicitly, or by using the indicator function. Preferably something that is easy to *swap* out for $L_l(x)$.
* "During training, we iterate over many samples of A* explorations which enlarges the scope of L ∗ ." - What does this mean? What is "many samples of A* explorations" exactly, is it over many different problem instances?
* Theorem 2 and proof: \bar should be over \Gamma, not h_\theta
* "the Iverson bracket is replaced by a logistics loss function $L_l(x) = $ ...": This should be clarified, see the first bullet point.
* "could solve 96% of levels of unfiltered boxoban mazes after seven epochs.": At epoch 8 then, which is not shown in Table 3? Please include it in Table 3.


Minor things
==========

* "The zero $L_2$ does not guarantee the optimal efficiency of A*" -> "$L_2$ = 0 does not ..."
* We propose a $L$* loss function" -> "We propose the $L$* loss function"
* A*-algorithm: "go to 4" -> "go to 5"?
* "Many prior art on" -> "Many prior works on"
* Footnote 3: $L_1$ and $L_2$ seem to be reversed in the second part of the sentence?
* "the extension to a set of plan is trivial": Would be clearer for readers if it is fleshed out a bit more.
* 4) "the problems identified above in Section 2.3": Be more specific here, e.g. what problems precisely (summarize).
* "DRC (3,3) allowed 1G iterations of SGD" and other "allowed" in the same sentence: "allowed" -> "used"?
* Table 3: Using a graph instead (i.e. $L_2$ and $L$* can use the same color for 3000, but one of them use line plot with '+' or similar) would give a better idea of rate of improvement, if shown as a function of epochs.

**Summary Of The Paper:**

The authors propose a novel loss function which is tailored specifically for learning A*-heuristics. $L$* captures (1) heuristic monotonicity and (2) optimal A*-search efficiency in terms of #node expansions. The authors motivates theoretically why $L$* overcome known limitations to L_p loss in terms of A* heuristics, and prove relevant theorems of $L$* to this end.

$L$* is evaluated on 3 difficult problems (Sokoban, mazes with teleports and Boxoban levels). $L$* is shown to consistently outperform $L_2$-loss as well as other state-of-the-art methods. Good generalization to problem instances with increased complexity not trained on is shown. Finally, a novel application of curriculum learning for A*-heuristics is demonstrated to be highly effective, both in terms of performance and the small number of SGD-steps necessary. Learning $L$* from random initialization and by only training on solved problems for every epoch, show an amazing performance increase at a small fraction of the cost in SGD-steps, in comparison with what is necessary for competing methods to reach similar performance.

**Summary Of The Review:**

The paper propose a simple but highly effective approach to learning good heuristics efficiently for A*-search in complex planning domains. The theoretical motivation is strong and well founded, the contributions significant, and the results well supported by a thorough empirical evaluation. If the demonstrated performance generalizes to other problem domains, then it is highly likely that the paper will have a high general impact.

---

### Official Review · Reviewer_SoyE · 2022-10-24

**Confidence:** 4
**Correctness:** 3
**Technical Novelty And Significance:** 3
**Empirical Novelty And Significance:** 2
**Recommendation:** 5

**Clarity, Quality, Novelty And Reproducibility:**

- The paper is mostly clear and easy to follow.

- The approach seems mostly novel, however there is missing literature (including techniques that do not rely on L_2 loss) that need to be cited and the difference and similarities to proposed approach should be highlighted. Details provided above.

- Reproducibility: the work seems reproducible and the author state that they will make "Scripts reproducing our
experiments together with mazes and solutions" available upon acceptance. However, I was not able to find the supplementary material that the paper mentioned should include the proofs and additional discussion.


**Strength And Weaknesses:**

Strengths:
* Approach is well motivated, highlighting problems with using L_2
* Experiments show improvement in coverage compared to the baselines

Weaknesses:
* Missing literature and baselines: there are many learning-based approaches for heuristic search that are not based on L_2 and are not cited in the paper [e.g., 1-4]. [1-2] have specifically focused on Sokoban. [3][4] are older works that avoid problems with L_2 by focusing on learning to rank.
* Optimality: the paper seems to focus on A* and is motivated by the "false sense of optimality" in L2, however the proposed approach is, to my understanding, not optimal. Specifically:
    - The theoretical optimality guarantees (Section 3.1) only hold for instances in the training set (i.e., to guarantee optimality in general, we would have to train on all possible instances).
     - L* is not differentiable and there is no guarantee that training find optimal solution (even with respect to the training set).
* Experimental results not sufficient to evaluate the proposed approach: Despite the focus on optimality, the experiments only focus on coverage and not on solution cost. Some of the baselines are not optimal, including Mercury 14, Stone soup, and the RL approach, as well as the proposed approach (to my understanding). It is therefore important to analyze what is the (average) obtained solution quality by each of the methods.
Given the lack of optimality and the comparison to non-optimal baselines, it would also be interesting to study the performance of the proposed L* heuristic function in greedy best-first search.
* Proofs are said to be in the supplementary material, however I could not find such material (neither at the end of the document nor uploaded to OpenReview).


[1] Orseau, Laurent, and Levi HS Lelis. "Policy-guided heuristic search with guarantees." Proceedings of the AAAI Conference on Artificial Intelligence. Vol. 35. No. 14. 2021.

[2] Feng, Dieqiao, Carla P. Gomes, and Bart Selman. "The Remarkable Effectiveness of Combining Policy and Value Networks in A*-based Deep RL for AI Planning." (2021).

[3] Garrett, Caelan Reed, Leslie Pack Kaelbling, and Tomás Lozano-Pérez. "Learning to rank for synthesizing planning heuristics." Proceedings of the Twenty-Fifth International Joint Conference on Artificial Intelligence. 2016.

[4] Xu, Yuehua, Alan Fern, and Sungwook Yoon. "Learning Linear Ranking Functions for Beam Search with Application to Planning." Journal of Machine Learning Research 10.7 (2009).

**Summary Of The Paper:**

The paper deals with the problem of learning heuristic function for A*, a popular heuristic search algorithm that can guarantee optimality of solutions and expands the minimal number of states. The paper proposes L*, a loss function tailored for A* which minimizes an upper bound on the number of expanded states. The paper motivates L* by highlighting the problem of using L_2 loss function as done in previous work and present experimental analysis that shows that heuristic functions learned by L* have better coverage (i.e., number of solved instances in time limit) compared to heuristic learned by L_2 loss and by other selected baselines.

**Summary Of The Review:**

Overall I think it is an interesting research direction and a well motivated approach that demonstrates nice experimental results. However, the paper is missing relevant literature, seem to have a conflict between the extensive focus on optimality and the actual theoretical guarantees the method provides, and lack of experimental results to experimentally support claims on optimality (i.e., analysis solution quality).

---

> ### Author Response · Authors · 2022-11-07
> **Clarification and supplementing length of optimal path**
>
> First of all, let us thanks for your valuable review.
>
> We admit to miss the works you have recommended, though we do not consider this to decrease the novelty of our work.  Levin Search Trees proposed in [1] and improved in [2] are very new and they propose a new search algorithm, whereas our work aims at fixing a very established approach to learn heuristic function for $A^*$ algorithm. According to our experimental results, $A^*$ algorithm with heuristic optimized by $L^*$ solves 100% instances of boxoban as search in [1], but unlike [1], the length of solution plans is very close to optimal (see results below). The ranking optimized in [3,4] certainly helps to remove problems with $L_2$ being symmetric, but it does not directly minimizes the number of expanded states and it does not solve the problem with multiple optimal path, which are also addressed by $L^*.$ We will of course happily add explanation of our contribution with respect to these missing works
>
> We fully agree with the reviewer that the optimal efficiency of $A^*$ can be assured only on the training set, or more generally on the  problem instances where the $L^*$ loss is zero on the optimal solution. We still believe this to be better then no guarantees provided by optimizing $L_2$ loss and its variants. The hope is that due to surprisingly good generalization of NNs, the performance would be good outside of the training set. Recall that in the evaluation presented in Table 1, the neural networks were trained on Sokoban mazes with 3 boxes yet they were able to solve problem instances with 9 boxes.
>
> While $L^*$ is not differentiable, the common approach is optimize convex surrogate. A rather old but important work [5,6] shows that by optimizing convex surrogate one can converge to optimal solution of the original problem. Hence, we do not see optimizing convex surrogate (in our case a logistic loss) as a problem. The theoretical properties assume non-differentiable 0-1 loss.
>
> We did not put the table with optimal solution length due to shortage of space. Below table shows an average length of plans solved by all methods, hence the numbers are comparable. The caption of columns are as in paper, but importantly SB* corresponds to Symba* optimal planner. We see that plans found by CoAT-L* are close to optimal, but we admit that according to our results, the CoAt architecture is more important than the loss function with respect to which the neural network is optimized. Yet, $A^*$ with NN optimizing $L^*$ delivers consistently shorter plans while solving more problem instances.
>
> n   | SB*   | Merc  | FDSS   |CNN-L2 | CNN-L*| CoAT-L2 | CoAT-L* |
>
> 3   | 21.40 | 29.32 | 27.89  | 30.56 | 28.67 | 22.90   | 22.02   |
>
> 4   | 34.00 | 41.00 | 37.00  | 43.42 | 41.33 | 35.11   | 35.03   |
>
> 5   | 38.82 | 45.76 | 42.37  | 45.34 | 44.83 | 40.12   | 40.12   |
>
> 6   | 41.11 | -------- | -------- |  49.82| 46.32 | 42.11   | 41.65   |
>
> 7   | -------- | --------  | -------- | 58.23 | 56.33 | 53.33   | 53.19   |
>
> The above experiments involving neural networks were repeated five times (including training), but the standard deviation was small between 0.004 and 0.008.
>
> We have performed the test of $L^*$ with GBF search and we will try to compile the results and post them below. But as has been discussed in detail above in answer to reviewer 8chY, $L^*$  is very tight to $A^*$ and it will perform sub-optimally to GBF search.
>
> We are sorry to hear that the supplemental material is missing, as it was prepared and it contained the above table with length of plans of different solvers.
>
> We wish our answer has cleared doubts and clarified the improvements over state of the art.
>
> References:
>
> [5] Steinwart, Ingo. "Consistency of support vector machines and other regularized kernel classifiers." IEEE transactions on information theory 51.1 (2005): 128-142.
>
> [6] Bartlett, Peter L., Michael I. Jordan, and Jon D. McAuliffe. "Convexity, classification, and risk bounds." Journal of the American Statistical Association 101.473 (2006): 138-156.

---

> > ### Author Response · Authors · 2022-11-18
> > **Additional experimental results**
> >
> > As requested by Reviewer oNqW, we added evaluation on five more domains with bootstrapped evaluation protocol, where neural networks are trained on problem instances the very same network has trained.
> > Due to the lack of space and minimizing the changes in the structure of the paper, the experimental results and details of experiments are in the supplementary material (we have double-checked it is correctly uploaded_
> >
> > We copy them from them for you convenience. In all cases, were report fraction of solved problem instances with respect to epoch, where epoch corresponds to a complete pass through problem instances, where the algorithm tries to solve yet unsolved.
> >
> > Sliding tile puzzle:
> >  |$L_*$ | $L_2$ |
> > 0 |   5 | 5 |
> > 1 |  11 | 8 |
> > 2 |  20 |18 |
> > 3 |  32 |22 |
> > 4 |  43 |30 |
> > 5 |  54 |39 |
> > 6 |  63 |42 |
> > 7 |  68 |54 |
> >
> >
> >    |  Blocks | | Ferry | | Gripper | | N-Puzzle
> >  ep| $L_*$ | $L_2$ | |$L_*$ | $L_2$ | |$L_*$ | $L_2$| |$L_*$ | $L_2$
> >  1 | 98  | 73  || 33 | 33 || 14 | 14 || 14 | 14
> >  2 | 100 | 75  || 48 | 44 || 15 | 15 || 52 | 34
> >  3 | 100 | 76  || 53 | 48 || 16 | 15 || 55 | 36
> >  4 | 100 | 76  || 54 | 49 || 17 | 16 || 55 | 38
> >  5 | 100 | 78  || 54 | 51 || 17 | 16 || 58 | 42
> >  6 | 100 | 78  || 54 | 51 || 17 | 16 || 58 | 43
> >  7 | 100 | 79  || 54 | 51 || 17 | 16 || 60 | 44
> >  8 | 100 | 80  || 54 | 52 || 18 | 16 || 60 | 44
> >  9 | 100 | 80  || 55 | 52 || 18 | 16 || 60 | 46
> > 10 | 100 | 80  || 55 | 52 || 19 | 16 || 62 | 46
> >
> > As you can see, in all cases, $L_*$ solves more problems that $L_2$.

---

> > > ### Comment · Reviewer_SoyE · 2022-12-11
> > > **Thank you for your response**
> > >
> > > I thank the authors for their response.
> > >
> > > The additional experiments alleviate concerns on the empirical performance of the proposed approach. I have increased my evaluation score.

---

### Official Review · Reviewer_8chY · 2022-10-25

**Confidence:** 4
**Correctness:** 3
**Technical Novelty And Significance:** 2
**Empirical Novelty And Significance:** 2
**Recommendation:** 5

**Clarity, Quality, Novelty And Reproducibility:**

I think the paper was well-written. To my knowledge, this is the first paper looking at different losses than L2 for heuristic search, so it is novel in that respect. It should be relatively reproducible.

**Strength And Weaknesses:**

I like the general direction of this paper. I could see how since L2 is not directly optimizing the main objective of being a good heuristic, a different objective could do better.
However, I think that the framing does not compare L* to the current state of the art.
In particular, one main paper that was not cited is DeepCubeA. In that paper, they use the L2 loss, but use value iteration and curriculum learning to get around the problems laid out with L2 in this paper. So I would really like to see comparisons to DeepCubeA, where the L2 loss is swapped out with L*.


**Summary Of The Paper:**

This paper proposes a new loss, called L* as opposed to L2 loss for training a heuristic to be used in A* search.


**Summary Of The Review:**

In summary, I am not convinced that L2 loss is bad. In particular, DeepCubeA was able to solve Sokoban and Rubik's cube using L2 loss. To be really convincing, I would like to see a problem where DeepCubeA fails because of the L2 loss, but the L* loss is able to solve the problem.

_____

After reading the comment I have bumped my score a bit but still recommend reject because I would like to see direct comparisons with DeepCubeA.

---

> ### Author Response · Authors · 2022-11-07
> **Clarification on optimality of L***
>
> First, let us thank you for your comments and let us clarify some misunderstandings.
>
> The optimality of $L^*$ stems from the fact that it directly minimizes the number of expanded states in the $A^*$ algorithm. This means that  if  $L^*$ is equal to zero on a problem instance, it is guaranteed that  $A^*$ algorithm will be on that problem instance optimally efficient. As such, $L^*$ is tied to $A^*$ so much, that it might perform sub-optimally with BFS. This is because $L^*$ assumes that $A^*$ uses $f = g + h$ as a heuristic and not just $h$ as in BFS, where $h$ denotes a heuristic value typically to be cost-to-goal (but not in $L^*$)  and $g$ is cost from initial state to the current one.
>
> One part of the problem with optimizing $h$ to estimate cost-to-goal by minimizing $L_2$ loss is that $L_2$ loss equally penalizes the over- and under-estimation of the true value. But in our view the bigger problem is that $h$ being cost-to-goal does not guarantee optimal efficiency of $A^*$ when problem instance contain multiple solution path with same cost (see [1] for an in-depth discussion), which can be confusing for $A^*$ if a tie-breaking algorithm is not provided. The proposed $L^*$ elegantly solves both problems, and furthermore solves the problem of assigning value to dead-ends.
>
> Now the value iteration algorithm used in DeepCubeA is designed to estimate cost-to-goal and as such it does not address the above discussed weakness of cost-to-goal heuristic in $A^*.$ Experimental comparison in our paper contains variants of CNNs and CoAt, where the estimated of cost-to-goal is optimized in supervised approach with respect to ground truth by minimizing the usual $L_2$ loss. We cannot see how value iteration would be better than direct optimization of ground truth values.
>
> Besides value iteration algorithm, DeepCubeA uses version of $A^*$, where computation of heuristics is performed in batches, which better utilizes modern HW (particularly GPU) and numerical libraries (BLAS, XLA, CuDNN). This improvement is therefore perpendicular to our work. It is important, but address a different inefficiency. In fact, we have discussed this improvement in past but left it as an interesting but future work once all other interesting ideas are evaluated. We also prefer to report results with vanilla $A^*$, as they better demonstrate the improvements brought by $L^*$  Finally, it is important to note that in our experiments, GPU was used only for training, while all testing was done on CPU to make the comparison fair to classical solvers, which do not use GPU. To conclude, the experimental results presented in Table 1 and Table 2 denoted by CNN-L2 and CoAT-L2 are very good substitute for DeepCubeA.
>
>
> [1] Malte Helmert and Gabriele Röger. How good is almost perfect? In AAAI, pages 944–949. AAAI Press, 2008.

---

### Official Review · Reviewer_oNqW · 2022-10-26

**Confidence:** 5
**Correctness:** 2
**Technical Novelty And Significance:** 3
**Empirical Novelty And Significance:** 2
**Recommendation:** 8

**Clarity, Quality, Novelty And Reproducibility:**

Clarity

The paper is mostly easy to understand. However, I do miss a more formal treatment with respect to A*'s optimality. I don't understand what it means to have a "false sense of optimality", for some definition of optimality.

Quality

While the idea is promising and interesting, the execution is lacking (see my comments above).

Novelty

The loss function is novel, to the best of my knowledge.

Reproducibility

The idea is simple enough (which is something very positive!) that should be easy for someone to reproduce the results of this paper.

**Strength And Weaknesses:**

Strength

This paper tackles one of the key weaknesses of learning heuristic functions to guide the A* search, which is the lack of a proper loss function. The L* loss seems to be a good step in the direction of having a loss function that is more correlated with the A* search effort.

Weaknesses

While the problem the paper tackles is important and the idea for solving this problem is promising, the paper has several important weaknesses that might prevent it from being published at this point.

1. The paper presents two types empirical evaluations. In the first, one uses classical planners to generate a training set from which a heuristic function is learned. In the second, the algorithm learns from the problems it is able to solve. The first setting is hardly interesting at all because it assumes the existence of a planner that is able to solve the problems. The second setting is the interesting one because it is general and it doesn't assume prior knowledge or a system for generating training data. I would have preferred an extensive evaluation on the second setting than the current mix of evaluations.
2. The evaluation is somewhat week because the system is evaluated only on grid-based puzzles (Sokoban and Maze with teleports). Instead of showing results on the setting where classical planners generate a training set, I would have preferred to see results on more domains.
3. The comparison of optimal classical planners with A* with a learned heuristic function is problematic as the planners are finding optimal solutions, while A* is finding suboptimal ones. Even if the solution costs are near optimal, the problems can be substantially easier if you lift the optimality requirement. MCTS and model-free methods aren't good baselines as they tend to be weak. MCTS suffers when planning on long horizons such as those of Sokoban (see [2]). Model-free methods are in disadvantage as they don't use the model to search, so one should expect that they perform rather poorly compared to search algorithms.
4. A more fair comparison would be with satisficing planners, which don't have the guarantee of finding optimal solutions. Another baseline that should be included is WA* with a learned heuristic function with the L2 loss. WA* is surprisingly effective in the learning setting (see [5]). Levin tree search (LTS) [1][2] uses a policy to guide its search and it defines a loss function that is an upper bound on the size of the tree. One can then learn a policy that minimizes this loss. This is exactly what is done in this paper, but for a policy. It is natural to wonder how A* with the L* loss compares to LTS.
5. The paper misses important citations. Whenever talking about optimality of A*, the paper seems to be referring to [3]. The procedure described in Section 5.3 is called Bootstrap [4].

References

[1] Orseau, L.; Lelis, L.; Lattimore, T.; and Weber, T. 2018. Single-Agent Policy Tree Search With Guarantees. In Advances in Neural Information Processing Systems 31, 3201–3211. Curran Associates, Inc.

[2] Orseau, Laurent and Levi H. S. Lelis. “Policy-Guided Heuristic Search with Guarantees.” AAAI (2021).

[3] Rina Dechter and Judea Pearl. 1985. Generalized best-first search strategies and the optimality of A*. J. ACM 32, 3 (July 1985), 505–536. https://doi.org/10.1145/3828.3830
3.

[4] Arfaee, Shahab & Zilles, Sandra & Holte, Robert. (2011). Learning heuristic functions for large state spaces. Artif. Intell.. 175. 2075-2098. 10.1016/j.artint.2011.08.001.

[5] Solving the Rubik's Cube with Deep Reinforcement Learning and Search. Forest Agostinelli*, Stephen McAleer*, Alexander Shmakov*, Pierre Baldi. Nature Machine Intelligence, Volume 1, 2019


**Summary Of The Paper:**

The paper introduces a novel loss function L* for learning heuristic functions that attempts to shrink the size of the the A* search tree. While previous methods for learning heuristic functions relied on minimizing the L2 loss, this paper argues that L2 loss doesn't necessarily reduce the search tree size. Empirical results of Sokoban-like puzzles show that L* might indeed reduce the A* search effort.

**Summary Of The Review:**

The paper introduces a novel and interesting loss function for learning heuristic functions for guiding the A* search. The results are somewhat preliminary and that is why I recommend rejection at this point. I do strongly encourage the authors to resubmit the paper if it is indeed rejected from this conference. The issues related to writing, lack of formalism, and missing important citations should all be easy to fix. The evaluation is the problem. The evaluation will be much stronger once L* is tested on more problem domains and compared with proper baselines.

---

> ### Author Response · Authors · 2022-11-07
> **Clarification of points raised by the reviewer**
>
> First, let us thank you for a detailed comments. We answer them in detail below and would like to clarify upon request.
>
> 1. We agree that the first scenario, where we use classical planner to generate training set on which the heuristic is optimized may sound useless. Keep in mind though that the training set contained examples of Sokoban mazes with three boxes and these mazes are therefore relatively easy, if tje number of boxes serves as a proxy for complexity. The testing set contains mazes with up to 9 boxes, where L* solved problem instances that classical planners were not able to solve (in the time limit). This should justify the setting a bit. Moreover, this scenario is very common in publication published on top venues and journals, see [6,7,8,9] below, therefore we conclude that it is an established benchmark in the community.
>
> 2. Grid based puzzles were used because the parametric heuristic function can be easily implemented by CNN. We do not consider this to be a problem, since our contribution is independent to the architecture of the heuristic function taking state as an input and being differentiable with respect to its parameters. The community seems to accept this and many methods are evaluated exclusively on grid domains, for example the papers [1,2,5] you have mentioned we failed to cite and [6,9,10]. We admit though that we would absolutely love to test our algorithm on other classical planning domains, but we simply do not know NN architecture accepting problem instances from PDDL planners.
>
> 3. and 4. Table 1 and Table 2 contain results for Mercury 14 and Fast Downward Stone Soup 2018 (FDSS), which are satisficing planners. You can see that they fail to solve mazes with 8 and 9 boxes at all, and the proposed L* solves almost two times more mazes with 7 boxes than FDSS. We included MCTS and DRC as a SOTA on Boxoban (we admit we were unaware of the work of Levin tree search (LTS), but our solution achieves 100% of solved mazes as LTS. But indeed a comparison on more difficult mazes would be very beneficial to see differences to LTS. We would like to take it to the follow-up work. While WA* might be surprisingly effective, it would still require to tune the weight factor somehow.
>
> 5. The missing citations and correction of terminology are rather trivial to fix.
>
> *We would also like to stress out that the proposed* $L^*$ *is taylored for* $A^*$ *algorithm and it solves the weakness of heuristics estimating cost-to-goal and these heuristics in our opinion should be the main competitor. As such, we believe the paper to improve on the state of the art.*
>
> References:
>
> [6] Guez, Arthur, et al. "Learning to search with mctsnets." International conference on machine learning. PMLR, 2018.
>
> [7] Toyer, Sam, et al. "Asnets: Deep learning for generalised planning." Journal of Artificial Intelligence Research 68 (2020): 1-68.
>
> [8] Ferber, Patrick, Malte Helmert, and Jörg Hoffmann. "Neural network heuristics for classical planning: A study of hyperparameter space." ECAI 2020. IOS Press, 2020. 2346-2353.
>
> [9] Groshev, Edward, et al. "Learning generalized reactive policies using deep neural networks." Twenty-Eighth International Conference on Automated Planning and Scheduling. 2018.
>
> [10] Agostinelli, Forest, et al. "Solving the Rubik’s cube with deep reinforcement learning and search." Nature Machine Intelligence 1.8 (2019): 356-363.

---

> > ### Comment · Reviewer_oNqW · 2022-11-10
> > **Thank you**
> >
> > Thank you for your reply!
> >
> > I am sorry, but I still find the setting where one uses an existing algorithm to generate training data as not very useful---even if others used the same setting. The interesting setting is where you extrapolate and improve your heuristic function as you solve more problems.
> >
> > Thanks for pointing out that you do have satisficing planners in the evaluation, this is certainly helpful, but classical planners are still very different from what you are doing. The apples-to-apples comparison we are missing is WA* trained with L2 loss. The weight-value in weighted A* is easy to tune. If you don't have access to enough computational resources, just use $w=2$. If you can afford running the experiment multiple times, pick something like $w \in \{1.5, 2.0, 2.5, 3.0\}$. Based on what is published in the literature (e.g., Agostinelli et al.), I would expect WA* with any of these weights to perform so much better than A* in the learning setting with L2 loss.

---

> > > ### Author Response · Authors · 2022-11-14
> > > **Comparison to weighted A***
> > >
> > > Per your request, we have performed the experiment, where heuristic was optimized by usual L2 loss and used inside WA* with w=2. We have performed full experiment with bootstrapping (Table 3 in the manuscript) for case, when the set of problem instances contained 3000 problems of Sokoban with 3 boxes.
> > >
> > > The fraction of solved after each epoch of training on solved mazes is shown below  (legend is at the bottom of this post)
> > >
> > > ep  | L2-A* | L2-WA* | L*-A*
> > > 0 | 7.4      | 7.4    | 7.4
> > > 1 | 11    | 13     | 11
> > > 2 | 15    | 20     | 18
> > > 3 | 29    | 27     | 33
> > > 4 | 34    | 36     | 62
> > > 5 | 62    | 58     | 75
> > > 6 | 67    | 65     | 77
> > > 7 | 63    | 67     | 76
> > >
> > > We can observe that while L2-WA* might have some advantage at early stages of bootstrapping, L* gives much better results when the  optimization progresses. This result should not be that surprising, since L* is optimizing heuristic to make it optimal inside A*, whereas WA* tries to patch sub-optimal heuristic.
> > >
> > > We are currently running additional experiments, where we take heuristic optimized with respect to L2 loss for 7 epochs and measure the fraction of solved mazes by WA* with different values of W and post them once we have them in hand. We wanted to post the above results when fresh.
> > >
> > > L2-A* means that L2 loss is optimized during training, A* is used for solving the problem
> > > L2-WA* means that L2 loss is optimized during training, WA* with w=2 is used for solving the problem
> > > L*-A* means that L* loss is optimized during training, A* is used for solving the problem

---

> > > > ### Comment · Reviewer_oNqW · 2022-11-14
> > > > **Increasing Recommendation**
> > > >
> > > > Thank you for running the WA* experiments. I have moved my recommendation from "marginally below the acceptance threshold" to "marginally above". I still think the experiments could be much stronger with a more diverse pool of domains and if they were focused on the setting where one doesn't need a solver to generate training data for the system.

---

> > > > > ### Author Response · Authors · 2022-11-17
> > > > > **Adding more problems**
> > > > >
> > > > > We have added more domains, though due to a structure of the paper and page limit, the results are in the supplementary. All experiments are in bootstrapped regime, where the network trains on problem instances that have been solved by A* with that particular network.
> > > > >
> > > > > At first, we added sliding tile puzzle. The experimental setting, including CoAt network architecture, were the same as in Section 5.2 In below table, you can find fraction of solved problem instances (in percents). The problem instances were downloaded from https://github.com/levilelis/h-levin/, though we have used just 5000 instances to save computational time. We can see that optimizing the heuristic with respect to $L_*$  leads to much better performance than with respect to $L_2.$
> > > > >
> > > > >  |$L_*$ | $L_2$ |
> > > > > 0 |   5 | 5 |
> > > > > 1 |  11 | 8 |
> > > > > 2 |  20 |18 |
> > > > > 3 |  32 |22 |
> > > > > 4 |  43 |30 |
> > > > > 5 |  54 |39 |
> > > > > 6 |  63 |42 |
> > > > > 7 |  68 |54 |
> > > > >
> > > > > Furthermore, we decided to go ahead and implemented a heuristic function for problems encoded in PDDL language. We represented the state as multi-graphs (which restricts the predicates to have arity at most two) and adapt a graph neural network to estimate heuristic from the state. We have also mildly adapted the bootstrap protocol, such that the network was optimized after every 32 solved problem instances. The optimization run for 10 epochs. The fraction of solved instances is shown in below table. We again see that optimizing the heuristic with respect to $L_*$  leads to much better performance than with respect to $L_2.$ We admit that we do not know, why Ferry and Gripper seems to be so challenging, we believe one of the culprits might be small number of problem instances. The experiment is in more detail described in supplementary.
> > > > >
> > > > >    |  Blocks | | Ferry | | Gripper | | N-Puzzle
> > > > >  ep| $L_*$ | $L_2$ | |$L_*$ | $L_2$ | |$L_*$ | $L_2$| |$L_*$ | $L_2$
> > > > >  1 | 98  | 73  || 33 | 33 || 14 | 14 || 14 | 14
> > > > >  2 | 100 | 75  || 48 | 44 || 15 | 15 || 52 | 34
> > > > >  3 | 100 | 76  || 53 | 48 || 16 | 15 || 55 | 36
> > > > >  4 | 100 | 76  || 54 | 49 || 17 | 16 || 55 | 38
> > > > >  5 | 100 | 78  || 54 | 51 || 17 | 16 || 58 | 42
> > > > >  6 | 100 | 78  || 54 | 51 || 17 | 16 || 58 | 43
> > > > >  7 | 100 | 79  || 54 | 51 || 17 | 16 || 60 | 44
> > > > >  8 | 100 | 80  || 54 | 52 || 18 | 16 || 60 | 44
> > > > >  9 | 100 | 80  || 55 | 52 || 18 | 16 || 60 | 46
> > > > > 10 | 100 | 80  || 55 | 52 || 19 | 16 || 62 | 46
> > > > >
> > > > >
> > > > > We hope that the above experiments are sufficient to demonstrate that theoretically correct  $L_*$ consistently outperform a widely used $L_2.$

---

> > > > > > ### Comment · Reviewer_8chY · 2022-12-06
> > > > > > **Missing Citations**
> > > > > >
> > > > > > The paper is still missing key citations.

---

> > > > > > > ### Author Response · Authors · 2022-12-08
> > > > > > > **Missing citations**
> > > > > > >
> > > > > > > Dear Reviewer,
> > > > > > >
> > > > > > > we have tried hard to add all citations, but we are limited by page limit constraints and we do not like citations without proper explanation, how it relate to our work. If you suggest which citations you consider crucial, we will try to add them, but that can happen at expense of removing the others, but we are happy to do this trade-off.
> > > > > > >
> > > > > > > May we ask what is your sentiment about the additional experiments you have asked for and we have performed?
> > > > > > > Do you think that the experimental comparison of L* to L2, which now contains seven domains (three of them on grid, four of them given by PDDL), and in all of which L* is clearly better is sufficient? Does the experiments sufficiently demonstrate that the theory behind L* is valid in practice?
> > > > > > >
> > > > > > > Thanks for answers in advance.

---

> > > > > > ### Comment · Reviewer_oNqW · 2022-12-12
> > > > > > **Thank you!**
> > > > > >
> > > > > > Thank you for including all these new results. I think the paper makes a much more compelling case now that is shows that the new loss function is able to produce better results than L2 on a larger diversity of domains. And I apologize for taking so long to reply. It is just a very busy time of the year.
> > > > > >
> > > > > > I am increasing my score once again. Thank you!

---

> > > > > > > ### Author Response · Authors · 2022-12-14
> > > > > > > **Thanks a lot.**
> > > > > > >
> > > > > > > Thank you very much. We appreciate it a lot.

---

> > ### Comment · Reviewer_oNqW · 2022-11-10
> > **Problem Domains**
> >
> > The issue I have with the domains used in the paper is that they are very similar to one another, there is almost no diversity in the choice of domains. You can still pick other domains for which a CNN will work fine and add to the diversity of domains you experiment with. For example, you could use Sliding-Tile puzzles, Pancake puzzles or Rubik's Cube. They are all grid-like puzzles, but different from the puzzles you used. Moreover, others have shown that CNNs can be used to learn effective heuristic functions for these domains.

---

> > > ### Author Response · Authors · 2022-11-14
> > > **Variaty of problems**
> > >
> > > I agree that we could do more problems on grid domains, but there are many published papers verifying the idea just on one domain, e.g. [1,6], (we have two), hence we concluded it should not be comparably worse to others.
> > > I would like to highlight that our comparison was done very rigorously especially with respect to not to give L* any advantage. Specifically, the advantage of L* is that from a single solved problem instance, she can learn (and therefore see) from states off the optimal path, because the L* loss is based on inequalities. Contrary L2 loss can learn only from states on the optimal path, since we need to know exact cost-to-goal. To remove this advantage, we have used classical solver to find solution length of states off the optimal path, such that L2 and L* have seen exactly the same number of states during training (despite L* has comparatively less information). Moreover, all experiments were repeated three times without any selection of models.

---

### Decision · Program_Chairs · 2023-01-20

**Decision:**

Reject

**Justification For Why Not Higher Score:**

There were very substantial concerns about the validation, including not a broad enough set of test problems.

**Justification For Why Not Lower Score:**

N/A

**Metareview: Summary, Strengths And Weaknesses:**

The reviewers agreed that the new loss function proposed in this paper is a novel and good idea.   There was some moderately substantial disagreement about the quality and and conclusiveness of the experimental results.

The major concerns were:
- An inconsistent use of "optimal".   A* and other planning algorithms are used to seek actual optimal solutions, but here due to lack of guaranteed admissibility of the heuristic, results are not necessarily optimal.  Also, experimental results compared both "optimal" and "satisficing" planners without clarity.  It seems that the authors are interested in finding cost-efficient, though not necessarily optimal solutions.  But most of the experimental results focused on planning efficiency not cost efficiency.  This needs to be made much clearer.
- A question of where the training data were to be obtained---it is somewhat less motivating if you have to be able to solve the very type of planning instances in training that you will eventually be testing on.
- A lack of comparison to closest competing approaches, such as Orseau et al or Feng et al
- Similarity of testing domains (grid worlds using a CNN to implement the learned heuristic) which might not be at all representative of the much larger space of planning problems.



**Summary Of Ac-Reviewer Meeting:**

The reviewers agreed that the main idea of the paper was novel, interesting, and potentially impactful.   There remained significant concerns about the strength of the experimental results as outlined above.